# Free Vibration Analysis of Smart Laminated Functionally Graded CNT Reinforced Composite Plates via New Four-Variable Refined Plate Theory

**DOI:** 10.3390/ma12223675

**Published:** 2019-11-07

**Authors:** Tran Huu Quoc, Tran Minh Tu, Vu Van Tham

**Affiliations:** Faculty of Industrial and Civil Engineering, National University of Civil Engineering, Hanoi 100000, Vietnam; tutm@nuce.edu.vn (T.M.T.); vuthamxd@gmail.com (V.V.T.)

**Keywords:** free vibration, four-variable refined plate theory, piezoelectric material, FG-CNTRC, laminated composite

## Abstract

This paper presents a new four-variable refined plate theory for free vibration analysis of laminated piezoelectric functionally graded carbon nanotube-reinforced composite plates (PFG-CNTRC). The present theory includes a parabolic distribution of transverse shear strain through the thickness and satisfies zero traction boundary conditions at both free surfaces of the plates. Thus, no shear correction factor is required. The distribution of carbon nanotubes across the thickness of each FG-CNT layer can be functionally graded or uniformly distributed. Additionally, the electric potential in piezoelectric layers is assumed to be quadratically distributed across the thickness. Equations of motion for PFG-CNTRC rectangular plates are derived using both Maxwell’s equation and Hamilton’s principle. Using the Navier technique, natural frequencies of the simply supported hybrid plate with closed circuit and open circuit of electrical boundary conditions are calculated. New parametric studies regarding the effect of the volume fraction, the CNTs distribution, the number of layers, CNT fiber orientation and thickness of the piezoelectric layer on the free vibration response of hybrid plates are performed.

## 1. Introduction

A novel class of functionally graded materials (FGM) titled functionally graded carbon nanotube-reinforced composite (FG-CNTRC) plates was first introduced by Shen [1]. Shen’s study revealed that the distribution of CNT had a remarkable influence on the mechanical behaviors of the FG-CNTR plates. Since then, static, dynamic, and buckling behaviors of FG-CNTRC structures have been studied and reported in the literature. Alibeigloo and Liew [2] studied the bending response of simply supported FG-CNTRC rectangular plate under thermo–mechanical loads by using the 3D theory of elasticity. Zhu et al. [3] presented a finite element model to study bending and free vibration responses of thin-to-moderately thick FG-CNTRC plates using the first shear deformation plate theory (FSDT). Lei et al. [4] gave the solution for static analysis of laminated FG-CNTRC plates using the element-free *k*p-Ritz method. Huang et al. [5] analyzed the bending and free vibration characteristics of antisymmetrically laminated FG-CNTRC plates using the FSDT and simple four-variable theory. The static, vibration and buckling responses of FG-CNTRC resting on elastic foundation were investigated by Wattanasakulpong [6] and Nguyen et al. [7]. Additionally, Shen et al. [8] analyzed the buckling and post-buckling behaviors of symmetrically distributed CNT-reinforced composite plate, including thermal effects. Next, Shen et al. [9] examined the buckling loads and post-buckling equilibrium paths of the CNTRC plates assuming properties of CNTs were temperature-dependent. Using a higher-order shear deformation plate theory (HSDT), the nonlinear free vibration behaviors of the FG-CNTRC plates with an elastic foundation in the thermal environment was investigated by Wang and Shen [10]. That study used the perturbation technique to solve the nonlinear equations of motion. Mehar et al. [11] investigated the static response of the FG-CNTRC doubly curved shell panel, in which the geometric nonlinear and thermal dependent properties of the individual constituents were considered. Using FSDT and piston theory in determining the aerodynamic pressure, Asadi et al. [12] analyzed aeroelastic flutter of FG-CNTRC beams under axial compression and supersonic airflow. These authors continue to study the aero-thermoelastic behaviors of supersonic FG-CNTRC plates taking to account thermal effects in [13].

There have been a limited number of studies related to electromechanical coupling analysis of laminated FG-CNTRC plates with surface-embedded or bonded piezoelectric layers. Using the 3D-theory, Alibeigloo investigated the bending behaviors of the piezoelectric FG-CNTRC (PFG-CNTRC) plates under the mechanical uniform load [14], thermal load, and electric field [2]. Rafiee et al. [15] investigated initial geometrical imperfections in the large amplitude dynamic stability of PFG-CNTRC plates under the simultaneous effect of thermal and electrical loadings. Setoodeh et al. [16] studied the free vibration characteristic of PFG-CNTRC spherical panels by differential quadrature method based on the HSDT. Using the Ritz method with Chebyshev polynomials, Kiani [17] analyzed the free vibration of the PFG-CNTRC plates with opened and closed circuits electrical boundary conditions. In Kiani’s research, the electric potential in the piezoelectric layers was assumed to be linearly distributed through the thickness of the plate. Wu et al. [18] presented a buckling analysis of an arbitrarily thick PFG-CNTRC plate subjected to in-plane compressive loads using unified formulation. Nguyen et al. [19] used the extended isogeometric method with non-uniform rational B-spline and the HSDT to investigate the dynamic response of PFG-CNTRC plates. In the study of Selim et al. [20], an element-free IMLS-Ritz model based on Reddy’s HSDT for the active vibration control of PFG-CNTRC plates was presented. Song et al. [21] used velocity feedback and linear quadratic regulator LQR methods to study active vibration control of PFG-CNTRC cylindrical shells with bonded piezoelectric patches. Zhang et al. [22] used a genetic algorithm to study shape control of FG-CNTRC rectangular plates bonded with piezoelectric patches acting as actuators and sensors.

HSDT [23,24,25,26,27,28] is often desirable for the design of composite structures since it yields more accurate results than the CPT (classical plate theory) and the FSDT. However, these HSDTs have computational costs because the equations of motions based on these HSDT are more. Therefore, simple HSDT must be developed. Recently, based on HSDT, Shimpi [29] developed a new plate theory that has only two unknown displacements, in which the transverse shear stress variation across the thickness is parabolic and equals zero on free surfaces. After that, several researchers introduced a class of four-variable refined plate theory by adding two in-plane displacements and separating the transverse displacements into the bending component and shear component. Meiche et al. [30] presented a new four-variable refined plate theory with hyperbolic shape function for buckling and vibration analysis of FGM sandwich plates. Thai and Vo [31] developed a new sinusoidal shear deformation theory to analyze static and dynamic behaviors of FG plates. Then, another sinusoidal shear deformation theory was also presented by Thai and Kim [32] to investigate the bending and free vibration response of FG plates. Daouadji et al. [33] presented the static analysis of FG plates using a new higher-order shear deformation model.

In present work, a new plate theory with four unknown displacements is presented for free vibration analysis of FG-CNTR plates with two piezoelectric layers bonded at the free surfaces. The electric potential in piezoelectric layers is assumed to be quadratic through the thickness. Navier solution is applied to solve the governing equation of simply supported rectangular plates to obtain the frequencies of the smart FG-CNTRC plates with closed and open circuit electrical conditions. The accuracy of the proposed plate theory is indicated by comparing the obtained natural frequencies with existing results in the literature. Several examples are carried out to show the effects of volume fraction and distribution type of CNTs, the number of layers, CNT fiber orientation, and thickness of piezoelectric layers on the natural frequencies of hybrid plates.

## 2. Laminated PFG-CNTRC Plates

A hybrid laminated FG-CNTRC plate with integrated piezoelectric lamina at top and bottom surfaces is depicted in Figure 1. Width, length, core thickness, and thickness of each piezoelectric layer of the plate are denoted by *a* and *b*, *h* and *h_p_*. Four types of CNT distribution across the thickness of each FG-CNT layer namely UD, FG-V, FG-O, and FG-X are also indicated in Figure 1.

The CNT volume fractions for each FG-CNTRC lamina are assumed as follows [3]:(1)$$\begin{array}{ll} V_{CNT}=V_{CNT}^{*}\;\;\;\;\;\;\;\;\;\;\;\;\;\; & \left(\mathrm{UD}\right)\\ V_{CNT}(z)=\left(1-\frac{2\left|z\right|}{h}\right)V_{CNT}^{*}\; & \left(\mathrm{FG}-O\right)\\ V_{CNT}(z)=\frac{4\left|z\right|}{h}V_{CNT}^{*}\; & \left(\mathrm{FG}-X\right)\;\;\;\;\\ V_{CNT}(z)=\left(1+\frac{2z}{h}\right)V_{CNT}^{*}\; & \left(\mathrm{FG}-V\right)\;\; \end{array}$$ where:(2)$$V_{CNT}^{*}=\frac{w_{CNT}}{w_{CNT}+\left(\rho^{CNT}/\rho^{m}\right)-\left(\rho^{CNT}/\rho^{m}\right)w_{CNT}}$$

The effective elastic properties of each FG-CNTRC lamina can be written as follows [3]:(3)$$\begin{array}{l} E_{11}=\eta_{1}V_{CNT}E_{11}^{CNT}+V_{m}E^{m}\;;\;\;\frac{\eta_{2}}{E_{22}}=\frac{V_{CNT}}{E_{22}^{CNT}}+\frac{V_{m}}{E^{m}}\;\;;\;\;\\ \frac{\eta_{3}}{G_{12}}=\frac{V_{CNT}}{G_{12}^{CNT}}+\frac{V_{m}}{G^{m}};\;v_{12}=V_{CNT}^{*}v_{12}^{CNT}+V_{m}v^{m}\;;\;\rho =V_{CNT}\rho^{CNT}+V_{m}\rho^{m}\;\;\; \end{array}$$ where $E_{11}^{CNT},E_{22}^{CNT},G_{12}^{CNT}$ and $E_{m}$, $G_{m}$ are Young’s moduli and shear modulus of CNT and isotropic matrix, respectively; $\eta_{1},\eta_{2}$, and $\eta_{3}$ are called efficiency parameters. $V_{CNT}$ and $V_{m}$ are the volume fractions of CNT and of matrix, respectively; the Poisson ratio and mass density of CNT/matrix are denoted as $v_{12}^{CNT},\rho^{CNT}$ and $v^{m},\rho^{m}$, respectively.

The linear constitutive relations for the FG-CNTRC core can be expressed as
(4)$$\left\{\begin{array}{c} \sigma_{x}^{k}\\ \sigma_{y}^{k}\\ \tau_{xy}^{k}\\ \tau_{yz}^{k}\\ \tau_{xz}^{k} \end{array}\right\}=\left[\begin{array}{ccccc} {\bar{Q}}_{11}^{k} & {\bar{Q}}_{12}^{k} & 0 & 0 & 0\\ {\bar{Q}}_{12}^{k} & {\bar{Q}}_{22}^{k} & 0 & 0 & 0\\ 0 & 0 & {\bar{Q}}_{66}^{k} & 0 & 0\\ 0 & 0 & 0 & {\bar{Q}}_{44}^{k} & 0\\ 0 & 0 & 0 & 0 & {\bar{Q}}_{55}^{k} \end{array}\right]\left\{\begin{array}{c} \varepsilon_{x}\\ \varepsilon_{y}\\ \gamma_{xy}\\ \gamma_{yz}\\ \gamma_{xz} \end{array}\right\}$$ where ${\bar{Q}}_{\mathrm{ij}}^{k}$ are the transformed elastic coefficients related to elastic coefficients in material coordinates $Q_{\mathrm{ij}}$ [34]:
(5)$$\begin{array}{cc} \; & {\bar{Q}}_{11}^{k}=Q_{11}^{k}\cos^{4}\theta^{k}+2\left(Q_{12}^{k}+2Q_{66}^{k}\right)\sin^{2}\theta^{k}\cos^{2}\theta^{k}+Q_{22}^{k}\sin^{4}\theta^{k}\\ \; & {\bar{Q}}_{12}^{k}=\left(Q_{11}^{k}+Q_{22}^{k}-4Q_{66}^{k}\right)\sin^{2}\theta^{k}\cos^{2}\theta^{k}+Q_{12}^{k}\left(\sin^{4}\theta^{k}+\cos^{4}\theta^{k}\right)\\ \; & {\bar{Q}}_{22}^{k}=Q_{11}^{k}\sin^{4}\theta^{k}+2\left(Q_{12}^{k}+2Q_{66}^{k}\right)\sin^{2}\theta^{k}\cos^{2}\theta^{k}+Q_{22}^{k}\cos^{4}\theta^{k}\\ \; & {\bar{Q}}_{66}^{k}=\left[Q_{11}^{k}+Q_{22}^{k}-2\left(Q_{12}^{k}+Q_{66}^{k}\right)\right]\sin^{2}\theta^{k}\cos^{2}\theta^{k}+Q_{66}^{k}\left(\sin^{4}\theta^{k}+\cos^{4}\theta^{k}\right)\\ \; & {\bar{Q}}_{44}^{k}=Q_{44}^{k}\cos^{2}\theta^{k}+Q_{55}^{k}\sin^{2}\theta^{k}\\ \; & {\bar{Q}}_{55}^{k}=Q_{44}^{k}\sin^{2}\theta^{k}+Q_{55}^{k}\cos^{2}\theta^{k} \end{array}$$

For each the CNT layer:(6)$$\begin{array}{l} Q_{11}^{k}=\frac{E_{11}}{1-\nu_{12}\nu_{21}};\;Q_{12}^{k}=\frac{\nu_{12}E_{22}}{1-\nu_{12}\nu_{21}};\;Q_{22}^{k}=\frac{E_{22}}{1-\nu_{12}\nu_{21}};\;\\ Q_{44}^{k}=G_{23}^{k};{\;Q}_{55}^{k}=G_{13}^{k};{\;Q}_{66}^{k}=G_{12}^{k} \end{array}$$

The constitutive relations for a piezoelectric material can be expressed as [35](7)$$\left\{\begin{array}{c} \sigma_{x}^{pk}\\ \sigma_{y}^{pk}\\ \tau_{xy}^{pk}\\ \tau_{yz}^{pk}\\ \tau_{xz}^{pk} \end{array}\right\}=\left[\begin{array}{ccccc} {\bar{C}}_{11}^{k} & {\bar{C}}_{12}^{k} & 0 & 0 & 0\\ {\bar{C}}_{12}^{k} & {\bar{C}}_{11}^{k} & 0 & 0 & 0\\ 0 & 0 & \frac{1}{2}\left({\bar{C}}_{11}^{k}-{\bar{C}}_{12}^{k}\right) & 0 & 0\\ 0 & 0 & 0 & {\bar{C}}_{55}^{k} & 0\\ 0 & 0 & 0 & 0 & {\bar{C}}_{55}^{k} \end{array}\right]\left\{\begin{array}{c} \varepsilon_{x}\\ \varepsilon_{y}\\ \gamma_{xy}\\ \gamma_{yz}\\ \gamma_{xz} \end{array}\right\}-\left[\begin{array}{ccc} 0 & 0 & {\bar{e}}_{31}^{k}\\ 0 & 0 & {\bar{e}}_{31}^{k}\\ 0 & 0 & 0\\ -e_{15}^{k} & 0 & 0\\ 0 & -e_{15}^{k} & 0 \end{array}\right]\left\{\begin{array}{c} E_{x}^{k}\\ E_{y}^{k}\\ E_{z}^{k} \end{array}\right\}$$
(8)$$\left\{\begin{array}{c} D_{x}^{k}\\ D_{y}^{k}\\ D_{z}^{k} \end{array}\right\}=\left[\begin{array}{ccccc} 0 & 0 & 0 & e_{15}^{k} & 0\\ 0 & 0 & 0 & 0 & e_{15}^{k}\\ {\bar{e}}_{31}^{k} & {\bar{e}}_{31}^{k} & 0 & 0 & 0 \end{array}\right]\left\{\begin{array}{c} \varepsilon_{x}\\ \varepsilon_{y}\\ \gamma_{xy}\\ \gamma_{xz}\\ \gamma_{yz} \end{array}\right\}+\left[\begin{array}{ccc} p_{11}^{k} & 0 & 0\\ 0 & p_{11}^{k} & 0\\ 0 & 0 & {\bar{p}}_{33}^{k} \end{array}\right]\left\{\begin{array}{c} E_{x}^{k}\\ E_{y}^{k}\\ E_{z}^{k} \end{array}\right\}$$

The elastic constants for the piezoelectric layer:(9)$$\begin{array}{l} {\bar{C}}_{11}^{k}=C_{11}^{k}-\frac{{\left(C_{13}^{k}\right)}^{2}}{C_{33}^{k}};\;{\bar{C}}_{12}^{k}=C_{12}^{k}-\frac{{\left(C_{13}^{k}\right)}^{2}}{C_{33}^{k}};\;{\bar{C}}_{55}^{k}=C_{55}^{k};\;\\ {\bar{e}}_{31}^{k}=e_{31}^{k}-\frac{C_{13}^{k}}{C_{33}^{k}}e_{33}^{k};\;{\bar{p}}_{33}^{k}=p_{33}^{k}+\frac{e_{33}^{2k}}{C_{33}^{k}} \end{array}$$ where $\left[C_{\mathrm{ij}}^{k}\right]$ is the elastic constants matrix of the piezoelectric lamina, $\left[p_{\mathrm{ij}}^{k}\right]$ is the dielectric permittivity matrix, $\left[e_{\mathrm{ij}}^{k}\right]$ is the electromechanical coupling matrix, $\left\{D^{k}\right\}$ is the electrical displacement, and $\left\{E^{k}\right\}$ is the electric field in the piezoelectric lamina.

## 3. Kinematic Equations

According to the four-variable refined plate theory [30,31,32,33], the displacement components at an arbitrary point in the hybrid panel can be expressed as follows:(10)$$\begin{array}{l} \bar{u}\left(x,y,z,t\right)=u(x,y,t)-z\frac{\partial w_{b}}{\partial x}-f\left(z\right)\frac{\partial w_{s}}{\partial x};\\ \bar{v}\left(x,y,z,t\right)=v(x,y,t)-z\frac{\partial w_{b}}{\partial y}-f\left(z\right)\frac{\partial w_{s}}{\partial y};\\ \bar{w}\left(x,y,z,t\right)=w_{b}(x,y,t)+w_{s}(x,y,t) \end{array}$$ where u, v are the displacements of the corresponding point on the reference surface of the plate along *x* and *y* axis, respectively; $w_{b}$ and $w_{s}$ are the bending and shear components of the transverse displacement, respectively; the shape function $f\left(z\right)$ represents the distribution of the transverse shear stresses and strains along the thickness.

By supposing the shape function $f\left(z\right)$ satisfies the free transverse shear stress conditions on the free surfaces of the plates, a class of refined plate theory was developed by various researchers as shown in Table 1:

In this study, a new shape function $f\left(z\right)$ is supposed as follows:(11)$$f\left(z\right)=z\left[-\frac{1}{8}+\frac{3}{2}{\left(\frac{z}{h}\right)}^{2}\right]$$

The linear strain-displacement relations are written as:(12)$$\begin{array}{l} \varepsilon_{x}=\frac{\partial u}{\partial x}-z\frac{\partial^{2}w_{b}}{\partial x^{2}}-f(z)\frac{\partial^{2}w_{s}}{\partial x^{2}};\\ \varepsilon_{y}=\frac{\partial v}{\partial y}-z\frac{\partial^{2}w_{b}}{\partial y^{2}}-f(z)\frac{\partial^{2}w_{s}}{\partial y^{2}};\\ \varepsilon_{xy}=\frac{\partial u}{\partial y}+\frac{\partial v}{\partial x}-2z\frac{\partial^{2}w_{b}}{\partial x\partial y}-2f(z)\frac{\partial^{2}w_{s}}{\partial x\partial y};\\ \gamma_{yz}=\left(1-f^{’}(z)\right)\frac{\partial w_{s}}{\partial y};\\ \gamma_{xz}=\left(1-f^{’}(z)\right)\frac{\partial w_{s}}{\partial x} \end{array}$$

The variation of electric potential through the thickness of the piezoelectric lamina was proposed by Wu et al. [36]:(13)$$\begin{array}{ll} \Phi^{t}(x,y,z,t)=\phi^{t}(x,y,t)\left[1-{\left(\frac{z-h/2-h_{p}/2}{h_{p}/2}\right)}^{2}\right]+{\bar{f}}_{1}(x,y,t)z+{\bar{f}}_{2}(x,y,t)\; & h/2\leq \;z\;\leq h/2{\;+\;h}_{p}\\ \Phi^{b}(x,y,z,t)=\phi^{b}(x,y,t)\left[1-{\left(\frac{-z-h/2-h_{p}/2}{h_{p}/2}\right)}^{2}\right]+{\bar{f}}_{3}(x,y,t)z+{\bar{f}}_{4}(x,y,t)\; & -h/2-h_{p}\leq \;z\;\leq -h/2 \end{array}$$ where the unknowns ${\bar{f}}_{1},{\bar{f}}_{2},{\bar{f}}_{3}$ and ${\bar{f}}_{4}$ can be obtained by satisfying the specific electrical boundary condition. In this study, two cases of electrical boundary conditions are considered. For the closed circuit condition, both major surfaces of the piezoelectric lamina are circuited:(14)$$\Phi (z=\pm \frac{h}{2})=0;\;\Phi (z=\pm (\frac{h}{2}+h_{p}))=0$$

On the other hand, when one surface is kept at zero voltage and the other is electrically insulated, for the open circuit condition, the electrical boundary conditions are

(15)$$\Phi (z=\pm \frac{h}{2})=0;\;D_{z}(z=\pm (\frac{h}{2}+h_{p}))=0$$

In addition, from electric potential function, the electric field can be derived as

(16)$$\vec{E}=-\vec{\nabla }\Phi$$

Substituting the expressions in Equations (13) and (8) into Equations (14) and (15) yields the electrical potential distribution for the closed circuit (C-circuit) as
(17)$$\begin{array}{ll} \Phi^{t}(x,y,z,t)=\phi^{t}(x,y,t)\left[1-{\left(\frac{z-h/2-h_{p}/2}{h_{p}/2}\right)}^{2}\right]\; & h/2\leq \;z\;\leq h/2{\;+\;h}_{p}\\ \Phi^{b}(x,y,z,t)=\phi^{b}(x,y,t)\left[1-{\left(\frac{-z-h/2-h_{p}/2}{h_{p}/2}\right)}^{2}\right]\; & -h/2-h_{p}\leq \;z\;\leq -h/2 \end{array}$$ and for open circuit (O-circuit) as
(18)$$\begin{array}{cc} \; & \Phi^{t}\left(x,y,z,t\right)=\phi^{t}\left(x,y,t\right)\left[1-{\left(\frac{z-h/2-h_{p}/2}{h_{p}/2}\right)}^{2}+\frac{4\left(z-h/2\right)}{h_{p}}\right]+\frac{{\bar{e}}_{31}}{{\bar{p}}_{33}}\left[u_{,x}+v_{,y}+\right.\\ \; & \left.\left(h/2+h_{p}\right)\left(w_{b,xx}+w_{b,yy}\right)+f\left(z\right)\left(w_{s,xx}+w_{s,yy}\right)\right]\left(z-h/2\right)\;\;h/2\leq \;z\;\leq h/2\;+\;h_{p}\\ \; & \Phi^{b}\left(x,y,z,t\right)=\phi^{b}\left(x,y,t\right)\left[1-{\left(\frac{-z-h/2-h_{p}/2}{h_{p}/2}\right)}^{2}+\frac{-4\left(z+h/2\right)}{h_{p}}\right]+\frac{{\bar{e}}_{31}}{{\bar{p}}_{33}}\left[u_{,x}+v_{,y}-\right.\\ \; & \left.\left(h/2+h_{p}\right)\left(w_{b,xx}+w_{b,yy}\right)+f\left(z\right)\left(w_{s,xx}+w_{s,yy}\right)\right]\left(z+h/2\right)\;\;-h/2-h_{p}\leq \;z\;\leq -h/2 \end{array}$$

## 4. Equations of Motion

Hamilton’s principle is used herein to derive the governing differential equations of motion for the free vibration problem. Without external forces, the principle can be stated as [37]
(19)$$\int_{t_{1}}^{t_{2}}\left(\delta U-\delta K\right)dt=\;0$$ in which *δU* is the variation of the strain energy of the plate and may be expressed as (20)$$\begin{array}{ll} \delta U & =\int_{A}^{}\int_{-h/2}^{h/2}\left(\sigma_{x}\delta \varepsilon_{x}+\sigma_{y}\delta \varepsilon_{y}+\tau_{xy}\delta \gamma_{xy}+\tau_{xz}\delta \gamma_{xz}+\tau_{yz}\delta \gamma_{yz}\right)dzdA\\ & =\int_{A}^{}\left\{N_{x}\frac{\partial \delta u}{\partial x}-M_{x}^{b}\frac{\partial^{2}\delta w_{b}}{\partial x^{2}}-M_{x}^{s}\frac{\partial^{2}\delta w_{s}}{\partial x^{2}}+N_{y}\frac{\partial \delta v}{\partial y}-M_{y}^{b}\frac{\partial^{2}\delta w_{b}}{\partial y^{2}}-M_{y}^{s}\frac{\partial^{2}\delta w_{s}}{\partial y^{2}}\right.\\ & \left.\;+N_{xy}\left(\frac{\partial \delta u}{\partial y}+\frac{\partial \delta v}{\partial x}\right)-2M_{xy}^{b}\frac{\partial^{2}\delta w_{b}}{\partial x\partial y}-2M_{xy}^{s}\frac{\partial^{2}\delta w_{s}}{\partial x\partial y}+Q_{xz}\frac{\partial \delta w_{s}}{\partial x}+Q_{yz}\frac{\partial \delta w_{s}}{\partial y}\right\}dA \end{array}$$ where *N*, *M*, and *Q* are stress resultants and defined by
(21)$$\left\{\begin{array}{ccc} N_{x}, & N_{y}, & N_{xy}\\ M_{x}^{b}, & M_{y}^{b}, & M_{xy}^{b}\\ M_{x}^{s}, & M_{y}^{s}, & M_{xy}^{s}, \end{array}\right\}=\sum_{k=1}^{N}\int_{h_{k}}^{h_{k+1}}\left(\sigma_{x}^{k},\sigma_{y}^{k},\tau_{xy}^{k}\right)\left\{\begin{array}{c} 1\\ z\\ f(z) \end{array}\right\}dz;$$
(22)$$\left(Q_{xz}^{s},Q_{yz}^{s}\right)=\sum_{k=1}^{N}\int_{h_{k}}^{h_{k+1}}\left(\tau_{xz}^{k},\tau_{yz}^{k}\right)g(z)dz$$ and *δK* is the variation of the kinetic energy of the plate and can be written as follows: (23)$$\begin{array}{ll} \delta K & =\int_{V}\left(\dot{u}\delta \dot{u}+\dot{v}\delta \dot{v}+\dot{w}\delta \dot{w}\right)\rho dAdz\\ & =\int_{A}\left\{{\bar{I}}_{0}\left(\dot{u}\delta \dot{u}+\dot{v}\delta \dot{v}+\left({\dot{w}}_{b}+{\dot{w}}_{s}\right)\delta \left({\dot{w}}_{b}+{\dot{w}}_{s}\right)\right)\right.\\ & -{\bar{I}}_{1}\left(\dot{u}\frac{\partial \delta {\dot{w}}_{b}}{\partial x}+\frac{\partial {\dot{w}}_{b}}{\partial x}\delta \dot{u}+\dot{v}\frac{\partial \delta {\dot{w}}_{b}}{\partial y}+\frac{\partial }{\partial y}\delta \dot{v}\right)\\ & -{\bar{I}}_{3}\left(\dot{u}\frac{\partial \delta {\dot{w}}_{s}}{\partial x}+\frac{\partial {\dot{w}}_{s}}{\partial x}\delta \dot{u}+\dot{v}\frac{\partial \delta {\dot{w}}_{s}}{\partial y}+\frac{\partial {\dot{w}}_{s}}{\partial y}\delta \dot{v}\right)\\ & +{\bar{I}}_{2}\left(\frac{\partial {\dot{w}}_{b}}{\partial x}\frac{\partial \delta {\dot{w}}_{b}}{\partial x}+\frac{\partial {\dot{w}}_{b}}{\partial y}\frac{\partial \delta {\dot{w}}_{b}}{\partial y}\right)\\ & +{\bar{I}}_{5}\left(\frac{\partial {\dot{w}}_{s}}{\partial x}\frac{\partial \delta {\dot{w}}_{s}}{\partial x}+\frac{\partial {\dot{w}}_{s}}{\partial y}\frac{\partial \delta {\dot{w}}_{s}}{\partial y}\right)\\ & \left.+{\bar{I}}_{4}\left(\frac{\partial {\dot{w}}_{b}}{\partial x}\frac{\partial \delta {\dot{w}}_{s}}{\partial x}+\frac{\partial {\dot{w}}_{s}}{\partial x}\frac{\partial \delta {\dot{w}}_{b}}{\partial x}+\frac{\partial {\dot{w}}_{b}}{\partial y}\frac{\partial \delta {\dot{w}}_{s}}{\partial y}+\frac{\partial {\dot{w}}_{s}}{\partial y}\frac{\partial \delta {\dot{w}}_{b}}{\partial y}\right)\right\}dA \end{array}$$ where mass moments $\left(I_{0},I_{1},I_{2},I_{3},I_{4},I_{5}\right)$ are defined by
(24)$$\left(I_{0},I_{1},I_{2},I_{3},I_{4},I_{5}\right)=\sum_{k=1}^{n}\int_{h_{k}}^{h_{k+1}}\left(1,z,z^{2},f(z),zf(z),f^{2}(z)\right)\rho^{\left(k\right)}dz$$

Substituting Equation (12) into Equation (7), then the obtained results into Equation (21), and combine with the relations in Equation (16), the stress resultants are obtained as follows:(25)$$\left\{\begin{array}{c} N\\ M^{b}\\ M^{s} \end{array}\right\}=\left[\begin{array}{ccc} A & B & B^{s}\\ B & D & D^{s}\\ B^{s} & D^{s} & H^{s} \end{array}\right]\left\{\begin{array}{c} \varepsilon \\ \kappa^{b}\\ \kappa^{s} \end{array}\right\}+\left\{\begin{array}{c} N^{p}\\ M^{bp}\\ M^{sp} \end{array}\right\};\;Q=A^{s}\gamma +Q^{p}$$ where (26)$$N=\left\{\begin{array}{l} N_{x}\\ N_{y}\\ N_{xy} \end{array}\right\};\;M^{b}=\left\{\begin{array}{l} M_{x}^{b}\\ M_{y}^{b}\\ M_{xy}^{b} \end{array}\right\};\;M^{s}=\left\{\begin{array}{l} M_{x}^{s}\\ M_{y}^{s}\\ M_{xy}^{s} \end{array}\right\};\;Q=\left\{\begin{array}{l} Q_{yz}\\ Q_{xz} \end{array}\right\}$$
(27)$$N^{p}=\left\{\begin{array}{l} N_{x}^{p}\\ N_{y}^{p}\\ N_{xy}^{p} \end{array}\right\};\;M^{bp}=\left\{\begin{array}{l} M_{x}^{bp}\\ M_{y}^{bp}\\ M_{xy}^{bp} \end{array}\right\};\;M^{sp}=\left\{\begin{array}{l} M_{x}^{sp}\\ M_{y}^{sp}\\ M_{xy}^{sp} \end{array}\right\};\;Q^{p}=\left\{\begin{array}{l} Q_{yz}^{p}\\ Q_{xz}^{p} \end{array}\right\}$$
(28)$$\begin{array}{l} \varepsilon ={\left\{\frac{\partial u}{\partial x},\frac{\partial v}{\partial y},\frac{\partial u}{\partial y}+\frac{\partial v}{\partial x}\right\}}^{T};\;\kappa^{b}={\left\{-\frac{\partial^{2}w_{b}}{\partial x^{2}};-\frac{\partial^{2}w_{b}}{\partial y^{2}};-2\frac{\partial^{2}w_{b}}{\partial x\partial y}\right\}}^{T};\;\\ \kappa^{s}={\left\{-\frac{\partial^{2}w_{s}}{\partial x^{2}};-\frac{\partial^{2}w_{s}}{\partial y^{2}};-2\frac{\partial^{2}w_{s}}{\partial x\partial y}\right\}}^{T};\;\gamma ={\left\{\frac{\partial w_{s}}{\partial x},\frac{\partial w_{s}}{\partial y}\right\}}^{T} \end{array}$$
(29)$$A=\left[\begin{array}{ccc} A_{11} & A_{12} & A_{16}\\ A_{12} & A_{22} & A_{26}\\ A_{16} & A_{26} & A_{66} \end{array}\right];\;B=\left[\begin{array}{ccc} B_{11} & B_{12} & B_{16}\\ B_{12} & B_{22} & B_{26}\\ B_{16} & B_{26} & B_{66} \end{array}\right];\;D=\left[\begin{array}{ccc} D_{11} & D_{12} & D_{16}\\ D_{12} & D_{22} & D_{26}\\ D_{16} & D_{26} & D_{66} \end{array}\right]$$
(30)$$\begin{array}{cc} \; & B^{s}=\left[\begin{array}{ccc} B_{11}^{s} & B_{12}^{s} & B_{16}^{s}\\ B_{12}^{s} & B_{22}^{s} & B_{26}^{s}\\ B_{16}^{s} & B_{26}^{s} & B_{66}^{s} \end{array}\right];\;D^{s}=\left[\begin{array}{ccc} D_{11}^{s} & D_{12}^{s} & D_{16}^{s}\\ D_{12}^{s} & D_{22}^{s} & D_{26}^{s}\\ D_{16}^{s} & D_{26}^{s} & D_{66}^{s} \end{array}\right];\;H^{s}=\left[\begin{array}{ccc} H_{11}^{s} & H_{12}^{s} & H_{16}^{s}\\ H_{12}^{s} & H_{22}^{s} & H_{26}^{s}\\ H_{16}^{s} & H_{26}^{s} & H_{66}^{s} \end{array}\right];\;\\ \; & A^{s}=\left[\begin{array}{cc} A_{44}^{s} & A_{45}^{s}\\ A_{54}^{s} & A_{55}^{s} \end{array}\right] \end{array}$$ in which (31)$$\begin{array}{l} \left(A_{ij},B_{ij},D_{ij},B_{ij}^{s},D_{ij}^{s},H_{ij}^{s})=\sum_{k=1}^{N}\int_{h_{k}}^{h_{k+1}}(1,z,z^{2},f(z),zf(z),{\left(f(z)\right)}^{2}){\left({\bar{Q}}_{ij}\right)}_{k}dz\;(i,j=1,2,6\right)\\ A_{ij}^{s}=\sum_{k=1}^{N}\int_{h_{k}}^{h_{k+1}}{\left[1-f’(z)\right]}^{2}{\left({\bar{Q}}_{ij}\right)}_{k}dz\;(i,j=4,5) \end{array}$$
and
(32)$$\begin{array}{l} \left\{\left\{\begin{array}{c} N_{x}^{p}\\ N_{y}^{p}\\ N_{xy}^{p} \end{array}\right\},\left\{\begin{array}{c} M_{x}^{pb}\\ M_{y}^{pb}\\ M_{xy}^{pb} \end{array}\right\},\left\{\begin{array}{c} M_{x}^{ps}\\ M_{y}^{ps}\\ M_{xy}^{ps} \end{array}\right\}\right\}=\sum_{k=1}^{n}\int_{h_{k}}^{h_{k+1}}{\left\{\begin{array}{c} \sigma_{x}^{p}\\ \sigma_{y}^{p}\\ \tau_{xy}^{p} \end{array}\right\}}^{\left(k\right)}\left(1,\;z,\;f(z)\right)dz\\ \left\{\begin{array}{l} Q_{yz}^{p}\\ Q_{xz}^{p} \end{array}\right\}=\sum_{k=1}^{n}\int_{h_{k}}^{h_{k+1}}{\left\{\begin{array}{l} \tau_{yz}^{p}\\ \tau_{xz}^{p} \end{array}\right\}}^{\left(k\right)}\left[1-f’(z)\right]dz; \end{array}$$

Substituting the expressions of *δU* and *δK* from Equations (21)–(26) into Equation (20) and after some mathematical manipulations, we obtain the equations of motion of the plate as follow:(33)$$\begin{array}{cc} \; & \delta u:\frac{\partial N_{x}}{\partial x}+\frac{\partial N_{xy}}{\partial y}={\bar{I}}_{0}\ddot{u}-{\bar{I}}_{1}\frac{\partial {\overset{..}{w}}_{b}}{\partial x}-{\bar{I}}_{3}\frac{\partial {\overset{..}{w}}_{s}}{\partial x}\;\;\\ \; & \delta v:\frac{\partial N_{y}}{\partial y}+\frac{\partial N_{xy}}{\partial x}={\bar{I}}_{0}\ddot{v}-{\bar{I}}_{1}\frac{\partial {\overset{..}{w}}_{b}}{\partial y}-{\bar{I}}_{3}\frac{\partial {\overset{..}{w}}_{s}}{\partial y}\;\;\;\\ \; & \delta w_{b}:\frac{\partial^{2}M_{x}^{b}}{\partial x^{2}}+2\frac{\partial^{2}M_{xy}^{b}}{\partial x\partial y}+\frac{\partial^{2}M_{y}^{b}}{\partial y^{2}}={\bar{I}}_{0}\left({\overset{..}{w}}_{b}+{\overset{..}{w}}_{s}\right)+{\bar{I}}_{1}\left(\frac{\partial \ddot{u}}{\partial x}+\frac{\partial \ddot{u}}{\partial y}\right)\\ \; & -{\bar{I}}_{2}\nabla^{2}{\overset{..}{w}}_{b}-{\bar{I}}_{4}\nabla^{2}{\overset{..}{w}}_{s}\\ \; & \delta w_{s}:\frac{\partial^{2}M_{x}^{s}}{\partial x^{2}}+2\frac{\partial^{2}M_{xy}^{s}}{\partial x\partial y}+\frac{\partial^{2}M_{y}^{s}}{\partial y^{2}}+\frac{\partial Q_{xz}^{s}}{\partial x}+\frac{\partial Q_{yz}^{s}}{\partial y}={\bar{I}}_{0}\left({\overset{..}{w}}_{b}+{\overset{..}{w}}_{s}\right)\\ \; & +{\bar{I}}_{3}\left(\frac{\partial \ddot{u}}{\partial x}+\frac{\partial \ddot{v}}{\partial y}\right)-{\bar{I}}_{4}\nabla^{2}{\overset{..}{w}}_{b}-{\bar{I}}_{5}\nabla^{2}{\overset{-}{w}}_{s} \end{array}$$

In addition, the electric potential in piezoelectric lamina must satisfy Maxwell’s equation:(34)$$\int_{h}^{h+h_{p}}\vec{\nabla }.\vec{D}dz+\int_{-h-h_{p}}^{-h}\vec{\nabla }.\vec{D}dz=\sum_{k=1}^{n_{pie}}\int_{h_{k}}^{h_{k+1}}\left(\frac{\partial D_{x}^{k}}{\partial x}+\frac{\partial D_{y}^{k}}{\partial y}+\frac{\partial D_{z}^{k}}{\partial z}\right)dz=0$$

## 5. Solution Procedures

In this study, two sets of simply supported boundary conditions (SSSS) are used to develop the Navier solutions for rectangular laminated plates and are shown in Table 2.

To satisfy the above boundary conditions, the following expansion displacements $\left(u,v,w_{b},w_{s}\right)$ are chosen as in Table 3:

where $u_{mn},v_{mn},w_{bmn},w_{smn}$ are unknown coefficients to be determined, c=cos, s=sin, *α= mπ/a*, *β = nπ/b*.

In addition, the electrostatic potential can be expanded as follows:(35)$$\phi (x,y,t)=\sum_{n=1}^{\infty }\sum_{m=1}^{\infty }\phi_{mn}e^{i\omega t}s\left(\alpha x\right)s\left(\beta y\right)$$

Substituting Equation (35) and the displacements in Table 3 into the equations of motion Equations (33) and (34), one obtains the analytical solution in the following matrix form:(36)$$\begin{array}{c} \begin{array}{c} \left\{\left[\begin{array}{ccccc} \chi_{11} & \chi_{12} & \chi_{13} & \chi_{14} & \chi_{15}\\ \chi_{12} & \chi_{22} & \chi_{23} & \chi_{24} & \chi_{25}\\ \chi_{13} & \chi_{23} & \chi_{33} & \chi_{34} & \chi_{35}\\ \chi_{14} & \chi_{24} & \chi_{34} & \chi_{44} & \chi_{45}\\ \chi_{15} & \chi_{25} & \chi_{35} & \chi_{45} & \chi_{55} \end{array}\right]-\omega^{2}\left[\begin{array}{ccccc} \psi_{11} & \psi_{12} & \psi_{13} & \psi_{14} & \psi_{15}\\ \psi_{12} & \psi_{22} & \psi_{23} & \psi_{24} & \psi_{25}\\ \psi_{13} & \psi_{23} & \psi_{33} & \psi_{34} & \psi_{35}\\ \psi_{14} & \psi_{24} & \psi_{34} & \psi_{44} & \psi_{45}\\ \psi_{15} & \psi_{25} & \psi_{35} & \psi_{45} & \psi_{55} \end{array}\right]\right\}\left\{\begin{array}{c} u_{mn}\\ v_{mn}\\ w_{mn}^{b}\\ \begin{array}{c} w_{mn}^{s}\\ \phi_{mn} \end{array} \end{array}\right\}=\left\{\begin{array}{c} 0\\ 0\\ 0\\ \begin{array}{c} 0\\ 0 \end{array} \end{array}\right\} \end{array} \end{array}$$ where the matrix elements of Equation (36) are given in the Appendix A.

## 6. Results and Discussions

In this section, several numerical results are carried out and discussed to verify the accuracy and efficiency of the proposed theory in free vibration analysis of simply supported laminated piezoelectric rectangular plates. Furthermore, the influence of volume fraction of CNTs, distribution of CNTs, number of the lamina, CNT fiber orientation, and thickness of piezoelectric lamina on the natural frequencies of laminated plates are also investigated in detail.

### 6.1. Comparison Studies

#### 6.1.1. Example 1

The non-dimensional natural frequencies $\bar{\omega }=\omega_{mn}h\sqrt{\rho /G}$ of simply supported isotropic square plate were performed and compared with the existing results in Table 4:

It is worth noting that the results obtained by Srinivas et al. [38] used CPT, FSDT, and exact solutions, whereas the work of Shimpi et al. [29] was implemented using a new FSDT. It is seen that all obtained frequencies are in good agreement with available results.

#### 6.1.2. Example 2

The second comparison study as follows:

The fundamental frequency of a square laminated PFG-CNTRC with piezoelectric lamina was calculated and compared with the results of K. Nguyen-Quang et al. [19] using an isogeometric approach. The plate had length *a = b* = 0.4 m, thickness *h* = 0.05*a*. Two continuous piezoelectric (PZT-5A) lamina of thickness *h_p_* = 0.1*h* were bonded to the top and bottom surfaces of the host. The material elastic properties for the matrix, CNT, and piezoelectric are listed in Table 5. 

The CNT efficiency parameters are shown in Table 6.

The comparision results are listed in Table 7.

It can be seen that the present results agree well with those acquired by the isogeometric approach [19] for different volume fractions of CNTs, distribution of CNTs, number of layers, CNT fiber orientation, and electrical condition, which indicates the accuracy and correctness of the present formulation and solution method.

### 6.2. Parametric Studies

After showing the accuracy of the present model, the following new results for free vibration of laminated FG-CNTRC plates integrated with piezoelectric layers were investigated. The material elastic properties for the matrix, CNT, and piezoelectric material are shown in Table 5 and Table 6.

#### 6.2.1. Effect of FG-CNT Parameters

Natural frequencies of anti-symmetric cross-ply and angle-ply laminated PFG-CNTRC (*a = b* = 0.4m; *a/h* = 20) are shown in Table 8 and Table 9, Table 10 and Table 11, respectively. It is observed from these tables that the FG-X plates had the highest value of frequency, whereas the FG-O plates had the lowest one. Therefore, it can be concluded that the type of CNT distribution has a remarkable influence on the stiffness of the plate. In detail, the CNTs distributed close to the upper and lower surfaces of each FG-CNTRC layer were more efficient than those distributed near the mid-plane of each FG-CNTRC layer in increasing the stiffness of the laminated PFG-CNTRC. Table 8 reveals that with the increase in the CNT volume fraction, the natural frequencies of the plates increased accordingly; these results are presented in more detail in Figure 2. Table 8 also shows that at the fixed value of the thickness ratio, the stiffness of the plate increased as the number layer of CNT increased. The effects of the width-to-thickness ratio on the natural frequencies of angle-ply laminated PFG-CNTRC plates are also presented in Table 9. As expected, the frequencies decrease with the increment of *a/h*. This is because the plates become thinner with the increment of *a/h*, and as the results, the stiffness of the plate decreased.

Figure 2 shows the fundamental frequencies of anti-symmetric angle-ply [*p/(θ/−θ)*_3_/p] laminated PFG-CNTRC plates versus the lamination angle (*a = b* = 0.4m; *a/h* = 20). It can be seen that the fundamental frequency increased with the increase in lamination angle *θ* from 0 to 45, and decreased with *θ* values from 45 to 90 for all four CNT distribution types and three CNT volume fractions. This is compatible with conclusions in previous studies in the literature. The previous conclusions regarding the CNT distribution type are confirmed. Noted that the plate with FG-X distribution type had the highest frequency, while with FG-O type had the lowest one.

#### 6.2.2. Effect of Electrical Condition

The natural frequency of laminated cross-ply FG-CNTRC plates (*a = b* = 0.4 m; *a/h* = 20; V^*^_CNT_ = 0.28) coupled with closed and open piezoelectric layers are shown in Table 8, Table 9, Table 10 and Table 11 with different inlet parameters: CNT volume fraction, CNT distribution type, number of layers, lamination angle, and width-to-thickness ratio. It is seen from these tables that the frequencies of the plates increased as the electrical boundary conditions changed from the closed circuit to the open circuit. Figure 3, once again, indicates that the FG-CNTRC plates coupled with the open circuit of piezoelectric layers had a greater stiffness than the FG-CNTRC plates coupled with the closed circuit of the piezoelectric layers. This may be because the open circuit converts electric potential to mechanical energy while the closed circuit does not.

#### 6.2.3. Effect of Piezoelectric Layer Thickness

The effect of piezoelectric layer thickness on the natural frequency of hybrid plates (*a = b* = 0.4 m; FG-X; [p/(−45/45)_3_/p]) for different CNT volume fraction and width-to-thickness was examined. For this purpose, the natural frequency increment δ between O-circuit and C-circuit electrical conditions is defined as:(37)$$\delta =\frac{\omega_{O-circuit}-\omega_{C-circuit}}{\omega_{C-circuit}}100\%$$

In Figure 4a,b, the effects of piezoelectric layer thickness on the natural frequency increment δ for different CNT volume fractions and different *a/h* ratio are depicted, respectively. It is found that the natural frequency increment δ had a higher value with a lower volume fraction of CNT and a larger *a/h* ratio. Furthermore, it can be seen when the *h_p_/h* ratio increased, the natural frequency increment δ increased. Accordingly, piezoelectric layer thickness had a greater effect on the natural frequency of an O-circuit piezoelectric coupled plate than that of a C-circuit.

Furthermore, the variations of the frequency parameter ω(Hz) are plotted in Figure 5a,b for the open circuit condition with different CNT volume fractions and different width-to-thickness ratios, respectively. These figures indicate that the natural frequency of the hybrid plate decreased by increasing the thickness of the piezoelectric layer from zeros to a specific value. After this value, the natural frequencies were increased by the incrementing of the piezoelectric layer in the cases of moderately thick plates but seem to be unchanged in cases of thin plates. It can be concluded that the piezoelectric effect is more effective in the case of thick plates rather than thin ones.

## 7. Conclusions

In summary, this paper shows our contribution to the development of a new four-variable refined plate theory for free vibration analysis of laminated PFG-CNTRC plates. The comparison studies show that the present theory is not only accurate but also efficient in predicting the free vibration responses of the plates. 

Our insight indicates that the natural frequency of the hybrid plates is strongly affected by the volume fraction of CNT and the distribution type of CNT in the matrix. FG-X CNTRC plate had the highest frequency, while the FG-O CNTRC plate had the smallest frequency regarding all inlet studied parameters. In addition, the lamination angles of CNT fiber and number of CNT lamina have a significant effect on the stiffness of the hybrid plate. Numerical results also revealed that the piezoelectric effect was more prominent in plates bonded with O-circuit piezoelectric lamina because, during vibration, the O-circuit converts electric potential to mechanical energy.

The present theory is accurate and efficient in solving free vibration behaviors of laminated FG-CNT reinforced composite plates with the piezoelectric layer and may be useful in the study of similar composite structures.

## Figures and Tables

**Figure 1 materials-12-03675-f001:**
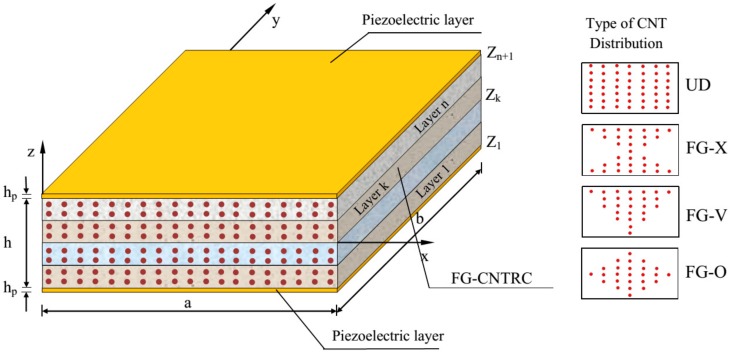
Configuration of the laminated piezoelectric functionally graded carbon nanotube-reinforced composite plates (PFG-CNTRC).

**Figure 2 materials-12-03675-f002:**
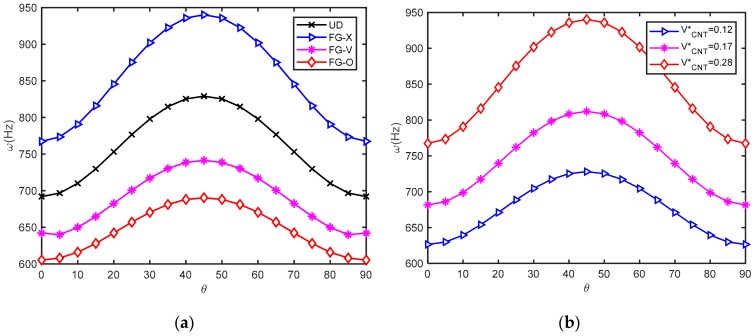
Effect of lamination angle on the natural frequency of laminated functionally graded carbon nanotube-reinforced composite plates (FG-CNTRC) plate coupled with O-circuit piezoelectric layer: (**a**) for different carbon nanotube (CNT) distribution types; (**b**) for different CNT volume fractions.

**Figure 3 materials-12-03675-f003:**
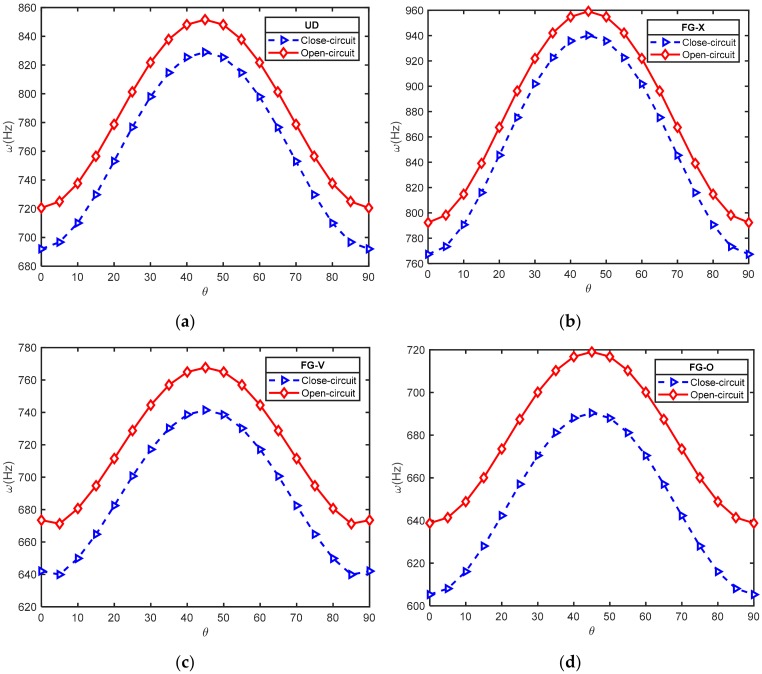
Effect of lamination angle on natural frequency of laminated FG-CNTRC plates [p/(−θ/θ)_3_/p] with electrical condition (*a = b* = 0.4m; *a/h* = 20; V^*^_CNT_ = 0.28): (**a**) UD; (**b**) FG-X; (**c**) FG-V; (**d**) FG-O.

**Figure 4 materials-12-03675-f004:**
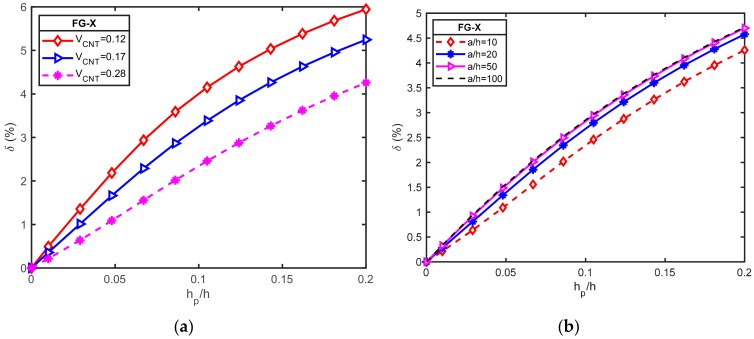
Variation of the natural frequency increment δ between O-circuit and C-circuit electrical conditions versus the *h_p_/h* ratio for a square piezoelectric functionally graded carbon nanotube-reinforced composite plates (PFG-CNTRC) plate (*a = b* = 0.4 m; FG-X; [p/(−45/45)_3_/p]): (**a**) for different CNT volume fractions; (**b**) for different *a/h* ratio.

**Figure 5 materials-12-03675-f005:**
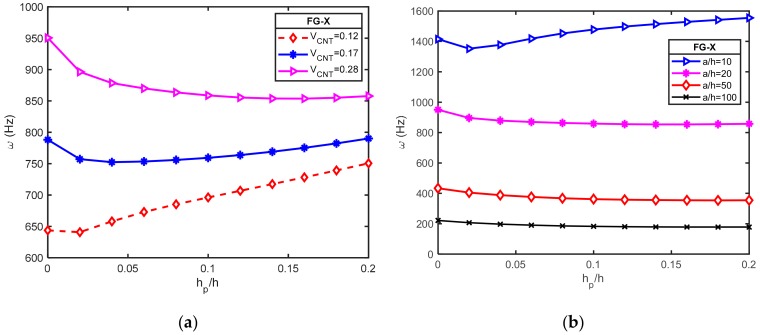
Effect of *h_p_/h* ratio on frequency parameter ω (Hz) of a square FG-CNTRC plate coupled with open circuit piezoelectric layer and different width-to-thickness ratio (*a = b* = 0.4m; V^*^_CNT_ = 0.28; [p/(−45/45)_3_/p]): (a) for different CNT volume fractions; (b) for different *a/h* ratio.

**Table 1 materials-12-03675-t001:** Shape functions of several four-variable refined plate theories.

Researcher	Shape Function
Shimpi [29]	$f\left(z\right)=z\left[-\frac{1}{4}+\frac{5}{3}{\left(\frac{z}{h}\right)}^{2}\right]$
N. E Meiche et al. [30]	$f\left(z\right)=\frac{\left(h/\pi \right)\sinh\left(\frac{\pi }{h}z\right)-z}{\left[\cosh\left(\pi /2\right)-1\right]}$
Huu-Tai Thai and Thuc P. Vo [31]	$f\left(z\right)=z-\frac{h}{\pi }\sin\frac{\pi z}{h}$
Huu-Tai Thai and Seung-Eock Kim [32]	$f\left(z\right)=\frac{4z^{3}}{3h^{2}}$
Daouadji et al. [33]	$f\left(z\right)=z-\left(z\sec h\left(\frac{\pi z^{2}}{h^{2}}\right)-z\sec h\left(\frac{\pi }{4}\right)\left(1-\frac{\pi }{2}\tanh\left(\frac{\pi }{4}\right)\right)\right)$

**Table 2 materials-12-03675-t002:** Two cases of simply supported boundary condition.

Edges	Boundary Conditions	
	Cross-Ply Laminates (SS-1)	Angle-Ply Laminates (SS-2)
*x* = 0 and *x* = a	$v=w_{b}=w_{s}=0;N_{x}=M_{x}^{b}=M_{x}^{s}=\;0$	$u=w_{b}=w_{s}=0;N_{x}=M_{x}^{b}=M_{x}^{s}=0$
*y* = 0 and *y* = b	$u=w_{b}=w_{s}=0;N_{y}=M_{y}^{b}=M_{y}^{s}=\;0$	$v=w_{b}=w_{s}=0;N_{y}=M_{y}^{b}=M_{y}^{s}=0$

**Table 3 materials-12-03675-t003:** The expansion displacements $\left(u,v,w_{b},w_{s}\right)$.

Displacements	Boundary Conditions	
	SS-1	SS-2
u(x,y,t)	$\sum_{m=1}^{\infty }\sum_{n=1}^{\infty }u_{mn}c\left(\alpha x\right)s\left(\beta y\right);\;$	$\sum_{m=1}^{\infty }\sum_{n=1}^{\infty }u_{mn}s\left(\alpha x\right)c\left(\beta y\right);\;$
v(x,y,t)	$\sum_{m=1}^{\infty }\sum_{n=1}^{\infty }v_{mn}s\left(\alpha x\right)c\left(\beta y\right);\;$	$\sum_{m=1}^{\infty }\sum_{n=1}^{\infty }v_{mn}c\left(\alpha x\right)s\left(\beta y\right);\;$
$w_{b}(x,y,t)$	$\sum_{m=1}^{\infty }\sum_{n=1}^{\infty }w_{mn}^{b}s\left(\alpha x\right)s\left(\beta y\right);$	$\sum_{m=1}^{\infty }\sum_{n=1}^{\infty }w_{mn}^{b}s\left(\alpha x\right)s\left(\beta y\right);\;$
$w_{s}(x,y,t)$	$\sum_{m=1}^{\infty }\sum_{n=1}^{\infty }w_{mn}^{s}s\left(\alpha x\right)s\left(\beta y\right);$	$\sum_{m=1}^{\infty }\sum_{n=1}^{\infty }w_{mn}^{s}s\left(\alpha x\right)s\left(\beta y\right);$

**Table 4 materials-12-03675-t004:** Non-dimensional natural frequencies $\bar{\omega }$ of simply supported boundary conditions (SSSS) isotropic square plate: a/h=10; b=a.

Mode		$\bar{\omega }$				
*m*	*n*	EXACT [38]	FSDT [38]	CPT [38]	Shimpi [29]	Present
1	1	0.0932	0.0930	0.0955	0.0930	0.0932
1	2	0.2226	0.2219	0.2360	0.2219	0.2232
2	2	0.3421	0.3406	0.3732	0.3406	0.3435
1	3	0.4171	0.4149	0.4629	0.4149	0.4192
2	3	0.5239	0.5206	0.5951	0.5206	0.5271
1	4	-	0.6520	0.7668	0.6520	0.6618
3	3	0.6889	0.6834	0.8090	0.6834	0.6941
2	4	0.7511	0.7446	0.8926	0.7447	0.7572
3	4	-	0.8896	1.0965	0.8897	0.9069
1	5	0.9268	0.9174	1.1365	0.9174	0.9356

**Table 5 materials-12-03675-t005:** Values of material parameters.

Core Plate		Piezoelectric Layer
CNT	Matrix	PZT-5A
*E*_11_^CNT^ = 5.64 TPa	*E_m_* = (3.52−0.0034T) (GPa)	*E* = 63 GPa; *G* = 23.3 GPa; *ν* = 0.35
*E*_22_^CNT^ = 7.0800 TPa	*ν*_m_ = 0.34	*ρ* = 7750 kg/m^3^
*G*_12_^CNT^ = 1.9455 TPa	*ρ*_m_ = 1150 kg/m^3^	e_31_ = −7.209 C/m^2^_,_ e_32_ = e_31_
*ν*_12_^CNT^ = 0.175		e_33_ = 15.118 C/m^2^
*ρ*^CNT^ = 1400 kg/m^3^		e_15_ = e_24_ = 12.322 C/m^2^
*G*_23_^CNT^ = 1.2 *G*_12_^CNT^		*p*_11_ = *p*_22_ = 1.53 × 10^−8^ F/m
		*p*_33_ = 1.5 × 10^−8^ F/m

**Table 6 materials-12-03675-t006:** Carbon nanotube (CNT) efficiency parameters with respect to various volume fractions.

V^*^_CNT_	*η* _1_	*η* _2_	*η* _3_
0.12	0.137	1.022	0.7*η*_2_
0.17	0.142	1.626	0.7*η*_2_
0.28	0.141	1.585	0.7*η*_2_

**Table 7 materials-12-03675-t007:** The fundamental natural frequency (Hz) of the SSSS square piezoelectric laminated piezoelectric functionally graded carbon nanotube-reinforced composite plates (PFG-CNTRC) (*a = b* = 0.4 m; *h_p_* = 0.1*h*; *a/h* = 20).

V^*^_CNT_	Type	Electrical Condition	Laminate Configurations					
			([p/0/p])		[p/0/90/0/p]		[p/(−45/45/−45)_as_/p]	
			Present	Ref. [19]	Present	Ref. [19]	Present	Ref. [19]
0.12	UD	C-circuit	587.099	583.199	587.099	583.510	662.579	656.538
	-	O-circuit	621.839	627.416	621.839	627.716	692.687	695.085
	FG-X	C-circuit	626.536	622.009	592.695	588.372	666.224	658.696
	-	O-circuit	658.751	662.982	627.080	632.184	696.144	697.103
	FG-V	C-circuit	563.624	560.042	585.314	581.714	661.328	655.606
	-	O-circuit	600.128	606.518	620.273	626.205	691.506	694.272
	FG-O	C-circuit	544.131	540.558	581.557	578.737	659.024	654.510
	-	O-circuit	581.965	588.764	616.659	623.343	689.323	693.196
0.17	UD	C-circuit	628.449	623.946	628.449	624.543	727.603	720.800
	-	O-circuit	660.700	665.032	660.700	665.615	754.615	755.388
	FG-X	C-circuit	681.622	675.814	636.195	631.317	732.516	723.781
	-	O-circuit	710.906	713.079	668.022	671.913	759.323	758.217
	FG-V	C-circuit	595.013	591.216	625.837	621.914	726.077	719.594
	-	O-circuit	629.510	635.182	658.370	663.359	753.169	754.324
	FG-O	C-circuit	569.202	565.533	620.976	618.126	723.043	718.247
	-	O-circuit	605.304	611.599	653.664	659.687	750.267	752.995
0.28	UD	C-circuit	692.023	685.587	692.023	686.852	828.991	821.713
	-	O-circuit	720.549	721.919	720.549	723.150	851.606	850.524
	FG-X	C-circuit	767.318	757.950	703.736	697.260	836.338	826.415
	-	O-circuit	792.364	789.814	731.760	732.991	858.755	855.093
	FG-V	C-circuit	642.030	637.353	688.175	682.974	827.574	820.463
	-	O-circuit	673.463	677.399	717.082	719.788	850.294	849.465
	FG-O	C-circuit	605.283	601.032	681.606	677.986	823.309	818.750
	-	O-circuit	638.738	643.745	710.669	714.904	846.167	847.767

**Table 8 materials-12-03675-t008:** The fundamental natural frequency ω(Hz) of anti-symmetric cross-ply [p/(0/90)_n_/p] laminated PFG-CNTRC plate (*a* = *b* = 0.4 m; *a/h* = 20; *h_p_* = 0.1*h*).

V^*^_cnt_	Type	Electrical Condition	Configuration			
			[p/(0/90)_1_/p]	[p/(0/90)_2_/p]	[p/(0/90)_3_/p]	[p/(0/90)_5_/p]
0.12	UD	C-circuit	535.019	574.472	581.514	585.093
	-	O-circuit	573.531	610.057	616.623	619.965
	FG-X	C-circuit	546.627	577.979	583.618	586.488
	-	O-circuit	584.256	613.333	618.591	621.270
	FG-V	C-circuit	530.609	573.512	581.131	584.998
	-	O-circuit	569.735	609.228	616.295	619.888
	FG-O	C-circuit	523.307	571.070	579.521	583.809
	-	O-circuit	562.746	606.888	614.765	618.768
0.17	UD	C-circuit	554.285	610.732	620.631	625.644
	-	O-circuit	591.418	644.038	653.340	658.059
	FG-X	C-circuit	570.880	615.687	623.633	627.665
	-	O-circuit	606.848	648.707	656.177	659.972
	FG-V	C-circuit	547.378	609.367	620.155	625.610
	-	O-circuit	585.362	642.842	652.935	658.048
	FG-O	C-circuit	537.761	606.167	618.011	623.996
	-	O-circuit	576.130	639.756	650.882	656.515
0.28	UD	C-circuit	575.055	664.853	680.081	687.747
	-	O-circuit	610.417	694.753	709.197	716.482
	FG-X	C-circuit	601.487	672.685	684.983	691.192
	-	O-circuit	635.165	702.232	713.903	719.805
	FG-V	C-circuit	563.213	663.041	679.760	688.157
	-	O-circuit	599.863	693.172	708.977	716.931
	FG-O	C-circuit	549.709	658.607	676.710	685.794
	-	O-circuit	586.877	688.865	706.029	714.662

**Table 9 materials-12-03675-t009:** The fundamental natural frequencies ω(Hz) of anti-symmetric angle-ply [p/(−θ/θ)_3_/p] laminated PFG-CNTRC plate (*a = b* = 0.4 m; *h_p_* = 0.1*h;* V^*^_CNT_ = 0.12).

Layers	Type	Electrical Condition	a/h			
			10	20	50	100
[p/(−5/5)_3_/p]	UD	C-circuit	1083.983	589.563	242.289	121.632
	-	O-circuit	1141.716	624.136	256.978	129.043
	FG-X	C-circuit	1087.691	591.770	243.224	122.104
	-	O-circuit	1145.139	626.204	257.859	129.488
	FG-V	C-circuit	1082.852	588.833	241.972	121.472
	-	O-circuit	1140.697	623.457	256.680	128.892
	FG-O	C-circuit	1080.548	587.458	241.389	121.177
	-	O-circuit	1138.576	622.171	256.131	128.615
[p/(−30/30)_3_/p]	UD	C-circuit	1166.673	644.809	266.550	133.932
	-	O-circuit	1217.212	675.925	279.922	140.691
	FG-X	C-circuit	1172.045	648.119	267.971	134.651
	-	O-circuit	1222.218	679.056	281.274	141.375
	FG-V	C-circuit	1164.971	643.678	266.053	133.680
	-	O-circuit	1215.655	674.862	279.450	140.452
	FG-O	C-circuit	1161.583	641.592	265.157	133.227
	-	O-circuit	1212.503	672.890	278.598	140.021
[p/(−45/45)_3_/p]	UD	C-circuit	1192.400	662.579	274.449	137.944
	-	O-circuit	1240.859	692.687	287.437	144.514
	FG-X	C-circuit	1198.251	666.224	276.021	138.740
	-	O-circuit	1246.327	696.144	288.937	145.273
	FG-V	C-circuit	1190.536	661.328	273.896	137.664
	-	O-circuit	1239.148	691.506	286.911	144.246
	FG-O	C-circuit	1186.834	659.024	272.903	137.162
	-	O-circuit	1235.694	689.323	285.965	143.767

**Table 10 materials-12-03675-t010:** The fundamental natural frequencies ω(Hz) of anti-symmetric angle-ply [p/(−θ/θ)_3_/p] laminated PFG-CNTRC plate (*a = b* = 0.4 m; *h_p_* = 0.1*h*; V^*^_CNT_ = 0.17).

Layers	Type	Electrical Condition	a/h			
			10	20	50	100
[p/(−5/5)_3_/p]	UD	C-circuit	1155.117	631.748	260.141	130.634
		O-circuit	1208.292	663.803	273.800	137.529
	FG-X	C-circuit	1160.475	634.869	261.455	131.296
		O-circuit	1213.327	666.758	275.047	138.157
	FG-V	C-circuit	1153.930	630.866	259.743	130.431
		O-circuit	1207.254	662.982	273.423	137.336
	FG-O	C-circuit	1150.769	628.983	258.944	130.028
		O-circuit	1204.308	661.204	272.665	136.954
[p/(−30/30)_3_/p]	UD	C-circuit	1261.941	704.587	292.376	146.997
		O-circuit	1306.976	732.703	304.536	153.149
	FG-X	C-circuit	1269.356	709.089	294.304	147.971
		O-circuit	1314.002	737.008	306.385	154.084
	FG-V	C-circuit	1260.134	703.205	291.744	146.674
		O-circuit	1305.348	731.399	303.931	152.840
	FG-O	C-circuit	1255.663	700.434	290.549	146.070
		O-circuit	1301.138	728.756	302.786	152.261
[p/(−45/45)_3_/p]	UD	C-circuit	1294.290	727.603	302.726	152.263
		O-circuit	1337.071	754.615	314.466	158.209
	FG-X	C-circuit	1302.277	732.516	304.840	153.332
		O-circuit	1344.658	759.323	316.500	159.237
	FG-V	C-circuit	1292.321	726.077	302.022	151.904
		O-circuit	1335.288	753.169	313.791	157.863
	FG-O	C-circuit	1287.487	723.043	300.707	151.238
		O-circuit	1330.721	750.267	312.527	157.223

**Table 11 materials-12-03675-t011:** The fundamental natural frequencies ω(Hz) of anti-symmetric angle-ply [p/(−θ/θ)_3_/p] laminated PFG-CNTRC plate (*a = b* = 0.4 m; *h_p_* = 0.1*h*; V^*^_CNT_ = 0.28).

Configuration	Type	Electrical Condition	*a/h*			
			10	20	50	100
[p/(−5/5)_3_/p]	UD	C-circuit	1252.438	696.727	288.711	145.122
		O-circuit	1298.062	725.018	300.910	151.292
	FG-X	C-circuit	1261.724	701.745	290.768	146.154
		O-circuit	1307.040	729.846	302.884	152.282
	FG-V	C-circuit	1252.602	695.985	288.272	144.891
		O-circuit	1298.480	724.357	300.493	151.071
	FG-O	C-circuit	1247.846	693.164	287.074	144.287
		O-circuit	1293.968	721.661	299.345	150.492
[p/(−30/30)_3_/p]	UD	C-circuit	1391.451	797.939	334.643	168.530
		O-circuit	1428.003	821.744	345.144	173.861
	FG-X	C-circuit	1403.564	804.759	337.484	169.959
		O-circuit	1439.786	828.365	347.899	175.247
	FG-V	C-circuit	1391.475	796.657	333.881	168.128
		O-circuit	1428.307	820.563	344.410	173.472
	FG-O	C-circuit	1385.151	792.713	332.170	167.262
		O-circuit	1422.249	816.756	342.754	172.633
[p/(−45/45)_3_/p]	UD	C-circuit	1431.534	828.991	349.087	175.920
		O-circuit	1465.742	851.606	359.140	181.030
	FG-X	C-circuit	1444.428	836.338	352.162	177.468
		O-circuit	1478.305	858.755	362.130	182.535
	FG-V	C-circuit	1431.592	827.574	348.231	175.467
		O-circuit	1466.081	850.294	358.314	180.591
	FG-O	C-circuit	1424.854	823.309	346.367	174.523
		O-circuit	1459.610	846.167	356.505	179.673

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
