# Peer review of "Free Vibration Analysis of Smart Laminated Functionally Graded CNT Reinforced Composite Plates via New Four-Variable Refined Plate Theory"

_materials, 2019, doi:10.3390/ma12223675_

Round 1
Reviewer 1 Report
In this paper, the author presented a new four-variable refined plate theory for free vibration analysis of functionally graded carbon nanotube reinforced laminated composite (FG-CNTRC) plate integrated with piezoelectric layers. This theory can solve free vibration behaviors of laminated FG-CNT reinforced composite plates with the piezoelectric layer.
The review has some questions.
First, the author presented four distribution types of CNT through the thickness of each FG-CNT layer. What do these UD, FG-V, FG-O and FG-X stand for? Are these CNT distribution types feasible to fabricate and how?
Second, the author listed several shape functions from preferences, so how did the author construct a new one in equation (11)?
Third, in the paper, the author used a lot of tables covering comparison study and parametric study. It is better to convert them into figure and it will be more straightforward and efficient to demonstrate the results.
Forth, the author should provide schematics for each situations.
Reviewer 2 Report
The paper is well written in terms of scientific background and content, however English syntax errors occur. The fact that readers conclude to remarks that are already well documented in literature poses a significant question of novelty. Authors are encouraged to revise the drawn results in such way. Additionally, the term smart is not evidenced throughout the text.
Author Response
Comment 1:
The paper is well written in terms of scientific background and content, however English syntax errors occur. The fact that readers conclude to remarks that are already well documented in literature poses a significant question of novelty. Authors are encouraged to revise the drawn results in such way. Additionally, the term smart is not evidenced throughout the text.
Response:
The manuscript has been proofread by a Vietnamese specialist and the English of the manuscript has been improved. We have converted data in Tables 11, 12 and 13 into graphs, and they are illustrated in Figures 4 and 5, and we hope that by this way the effectiveness of the proposed model will be clearer (all changes are highlighted in green). The term "smart" often refers to piezoelectric materials.

Reviewer 3 Report
Technically this manuscript is nice and properly written so I recommend to accept this manuscript
Author Response
Thank you so much!
Reviewer 4 Report
There are several concerns regarding this submission.
1) First of all, the significant part of literature has been ignored. Authors should add some recent papers on this topic such as:
*) Composites Part B, Volume 116, 1 May 2017, Pages 486-499
*) Composite Structures, Volume 171, 1 July 2017, Pages 113-125
2) This paper assumes perfectly straight and aligned CNTs inside the matrix which technically is impossible to be achieved in reality. SEM images from nanocomposites reveal curly geometry of CNTs which usually form bundles. The authors should comment on how their method can be extended so as to account for random geometry and distribution of CNTs inside the polymer and also for imperfect adhesion between the CNT and the polymer.
3) SWCNTs are selected as reinforcement in the present study. The authors should clarify that why is SWCNT used as reinforcement? It is better if the authors compare the effect of SWCNT with DWCNT or MWCNT or provide some justifications in this regard.
4) Interface between the CNTs and matrix is crucial in the mechanical behavior of the Nano composite. This element should be considered when calculating the material properties of the Nano composite. It seems that the authors ignore this physical phenomenon in nanoscale and obtained the efficiency parameters according to some oversimplified assumptions.
All in all, this submission needs major revision.
Author Response
Comments and Suggestions for Authors
There are several concerns regarding this submission.
1) First of all, the significant part of literature has been ignored. Authors should add some recent papers on this topic such as:
*) Composites Part B, Volume 116, 1 May 2017, Pages 486-499
*) Composite Structures, Volume 171, 1 July 2017, Pages 113-125
Response
Thank you for your advice, we have added these references to the Introduction part of the manuscript.
2) This paper assumes perfectly straight and aligned CNTs inside the matrix which technically is impossible to be achieved in reality. SEM images from nanocomposites reveal curly geometry of CNTs which usually form bundles. The authors should comment on how their method can be extended so as to account for random geometry and distribution of CNTs inside the polymer and also for imperfect adhesion between the CNT and the polymer.
Thank you for the very interesting comments. Regarding the technology of producing Carbon Nanotube Reinforced composite materials, there are many studies involving specific techniques. The technology and manufacturing processes for these materials are complex and are often of interest to experts in this field. Using a powder metallurgy fabrication process, carbon-nanotube reinforced composites (CNTRCs) may be achieved with a non-uniform distribution of CNTs through the media. This type of reinforced composite media is known as functionally graded carbon-nanotube-reinforced composite (FG-CNTRC). This new type of FG-CNT reinforced composite will need further research so as to find out its mechanical properties.
We only specialize in mechanics and the mechanical behavior of materials and structures, so the technology of producing this material is not our forte. Therefore, we hope that the reviewer can review and accepts this explanation and the assumptions used in our study.
3) SWCNTs are selected as reinforcement in the present study. The authors should clarify that why is SWCNT used as reinforcement? It is better if the authors compare the effect of SWCNT with DWCNT or MWCNT or provide some justifications in this regard.
4) Interface between the CNTs and matrix is crucial in the mechanical behavior of the Nano composite. This element should be considered when calculating the material properties of the Nano composite. It seems that the authors ignore this physical phenomenon in nanoscale and obtained the efficiency parameters according to some oversimplified assumptions.
Response
Thanks for the valuable comments. To answer this question thoroughly is difficult, because this is not our intensive research area. We choose SWCNTs as reinforcement in our study according to the scenario of previous studies that were assumed by many authors. However, we have consulted open sources and would like to answer briefly to our understanding as follows:
Amongst the multi-functional materials, carbon nanotubes (CNTs) are extensively applied to high performance composite structures in light of their extraordinary stiffness and strength, distinguished electrical and thermal properties [1,2]. Functionally graded carbon nanotube reinforced (FG-CNTRC) composites are prepared through a variety of processing techniques. Powder metallurgy is the most popular and widely applied technique for preparing CNT Reinforced FG composites. Based on the number of graphene layers forming a tube, carbon nanotubes can be classified as single-walled or multi-walled carbon nanotubes (double-walled carbon nanotubes). Experimental and numerical studies show that the performance of such composites depends critically on the CNT/matrix interfacial characteristics. Interface strength and interface length are two of the most important factors that affect the mechanical properties of CNT-reinforced composites [4].
Some works have attributed the differences between the tensile and compression strain cases to the sliding of inner shells of the MWNTs when a tensile stress is applied. In the cases of SWNT epoxy composites, the possible sliding of individual tubes in the SWCNT rope, which is bonded by van der Waals forces, may also reduce the efficiency of the load transfer. It is suggested that for the SWNT rope case, interlocking using epoxy molecules might bond the SWCNT rope more strongly [3].
Thus, it can be seen that the studies on the synthesis and processing of CNT-reinforced composites have attracted the attention of many experts. Many works in this area have been published. However, because we are only specialized in the field of mechanics, the details of the manufacturing process are idealized and the influence of factors related to manufacturing technology is ignored.
We hope that the reviewer will accept our explanation from the viewpoint that we are not experts in the manufacturing technology of materials.
[1] Asadi, H., Souri, M., & Wang, Q. (2017). A numerical study on flow-induced instabilities of supersonic FG-CNT reinforced composite flat panels in thermal environments. Composite Structures, 171, 113-125.
[2] Asadi, H., & Wang, Q. (2017). An investigation on the aeroelastic flutter characteristics of FG-CNTRC beams in the supersonic flow. Composites Part B: Engineering, 116, 486-499.
[3] Randjbaran, E., Zahari, R., Majid, D. L., Sultan, M. T., & Mazlan, N. (2017). Reasons of adding carbon nanotubes into composite systems-review paper. Mechanics and Mechanical Engineering, 21(3).
[4] Loos, Marcio. Carbon nanotube reinforced composites: CNT Polymer Science and Technology. Elsevier, 2014.

Round 2
Reviewer 1 Report
The authors addressed all comments from the reviewer. This manuscript can be accepted in the present form.
Author Response
Thank you so much!